# An Optoelectronic Detector with High Precision for Compact Grating Encoder Application

Yusong Mu [1], Nanjian Hou [1], Chao Wang [1], Yang Zhao [1], Kaixin Chen [2] and Yaodan Chi [1,*]

1 Key Laboratory for Comprehensive Energy Saving of Cold Regions Architecture of Ministry of Education, College of Electronic and Computer, Jilin Jianzhu University, Changchun 130118, China
2 Northeast Electric Power Design Institute Co., Ltd. of China Power Engineering Consulting Group, Changchun 130000, China
* Correspondence: chiyaodan@jlju.edu.cn; Tel.: +86-135-0446-4607

**Abstract:** This paper presents a novel optoelectronic detection array that adopts the research idea of optical, mechanical and electrical integration. Through the design of new detectors and ASIC, the mutual restriction between high accuracy and miniaturization of the grating encoder is solved. A simulation model of the "broken line" detector structure and process was established that only meets the needs of a compact array layout but also ensures a good photoelectric conversion rate. In addition, we used a professional design program to complete the layout of the ASIC, which maximized the recovery of the signal received by the detector. The simulation and noise analysis results show that the SNRs of the output signal are greater than 60 dB with a 400 kHz response frequency.

**Keywords:** optoelectronic detector; integrated circuit; phase difference filtering; noise optimization

## 1. Introduction

As a kind of precision angle measuring device with circular grating as a measuring element, the encoder can be divided into two categories according to its use. One is high-precision products used in astronomy, military industry, precision instruments and other fields. The representative is the incremental and absolute hybrid encoder designed by the Heidenhain company of Germany for the Galileo (Italy) telescope control system. Its accuracy is 0.036" RMS, which is one of the encoders with the highest accuracy at present [1,2]. The other is miniaturized products used in the industrial field with 1" RMS. In order to better meet the needs of applications, such products have been designed in a compromised way on system accuracy and system miniaturization. Therefore, system accuracy is the most important performance index for any kind of encoder [3–5].

In recent years, researchers have studied the optical and mechanical structure, coding methods, signal processing and other aspects of the encoder in order to improve the accuracy. Kim et al. [6] developed a new absolute angle measurement method that uses the 10-bit and 13-bit PE-BGD, where the APBC can be sub-divided by calculating the relative position between the binary patterns and the detector pixel, and suggested that a specially designed photodiode array was used to obtain a higher signal-to-noise ratio and measurement speed. Thereafter, Yu et al. [7] designed a photoelectric encoder that is small in size while ensuring it had sufficiently high resolution and accuracy, and Jiang et al. [8] designed a phased array optoelectronic detector using phase difference filtering technology for Incremental Encoder Application. The research results presented here may provide a theoretical and technological foundation for further research on small-size, high-resolution photographic rotary encoders. At the same time, moiré fringe signal error compensation technology has been a research hotspot in this field, so algorithm compensation, optical compensation and many other compensation methods have been proposed. Hou et al. [9] realized high-precision measurement technology of a single-excitation petal-shaped capacitive encoder based on periodic nonlinear error compensation. In 2020, Zhu et al. [10]

proposed a method to directly subdivide non-orthogonal moiré signals, which avoids the problem of high resource occupation caused by theoretical error introduced by the CORDIC algorithm in inverse cosine calculation and orthogonal error compensation. In 2022, Hou et al. [11] analyzed the generation principle of the grating moiré fringe subdivision error by the mathematical model to the characteristics of the grating signal waveform equation and proved through experiments that the method can make the overall system obtain better convergence efficiency and a more accurate fitness value. These studies have achieved success under experimental conditions, but they still face new challenges in industrial application and industrialization.

To sum up, we have adopted the research idea of optical, mechanical and electrical integration and look forward to integrating the phase difference filtering technology with the photodetector. I introduced a novel phased array optoelectronic detection device and its readout circuit in this manuscript. Its main feature is that the detector is designed in the shape of a broken line. The advantage of this feature is that the phase filter difference principle is used to weaken the influence of harmonics, and the output signal during photoelectric conversion is more sinusoidal. In order to achieve this feature, I verified the process feasibility of the broken line detector and the device performance of the detector through mathematical model simulation. In addition, I designed and implemented a special amplifying readout circuit, which has the advantages of low noise, high gain and high SNDR and simulation analysis of the process angle and noise of the layout. The specific application of the detector and its performance requirements mainly focus on the sinusoidal of the received photoelectric signal and the signal-to-noise ratio of the output signal. Therefore, the performance of other aspects of the chip is not over-designed to make sure to meet the application requirement while at minimum cost.

## 2. Design of the Phased Array Detector Chip

As shown in Figure 1, the grating encoder is a comprehensive system integrating light, machinery and electricity. In particular, the matching between the optical system and the photoelectric devices greatly affects the overall performance [12–15]. We specially develop photoelectric devices to obtain better signal quality, and such chips are very suitable for the design of integrated circuits for special applications. In this study, the detector is implemented on an array structure with an identical shape and size of the light window of the disk. This scheme changes the shape of the detector through the principle of phase difference filtering so as to weaken the harmonic. Compared with the traditional "moiré fringe" encoder structure [16,17], the scheme not only reduces the complexity of the optical system but also reduces the diffraction of primary light and ultimately improves the quality of the signal and system reliability. Further, integration of the detector and circuit is realized based on the X-FAB process. The key core technical indicators of the ASIC reach the highest level of similar products.

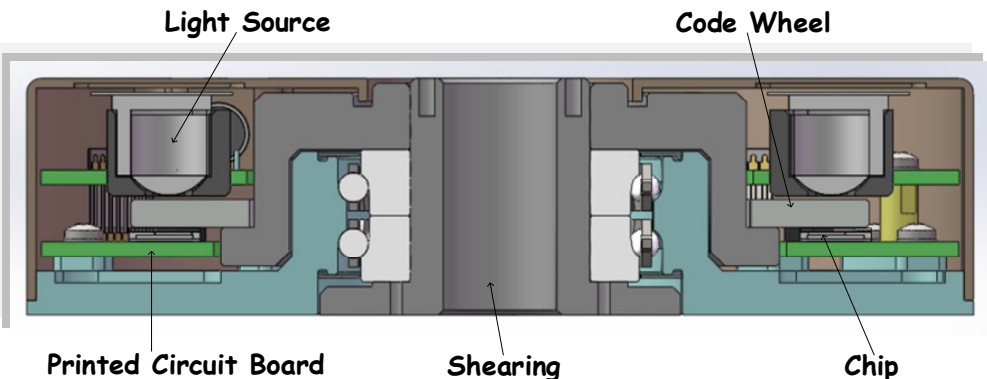

**Figure 1.** Phased array optoelectronic encoder structures.

The chip researched is a system-level chip specially used for grating encoders, mainly including a detector array and integrated circuit system. As shown in Figure 2, the detector array uses a phase array to receive optical information and then converts the received optical signal into a current signal. SoC is responsible for converting the weak current output by the detector into a voltage signal and outputting it outside the chip. The chip adopts the design idea of combining a photodetector with a processing circuit [18,19].

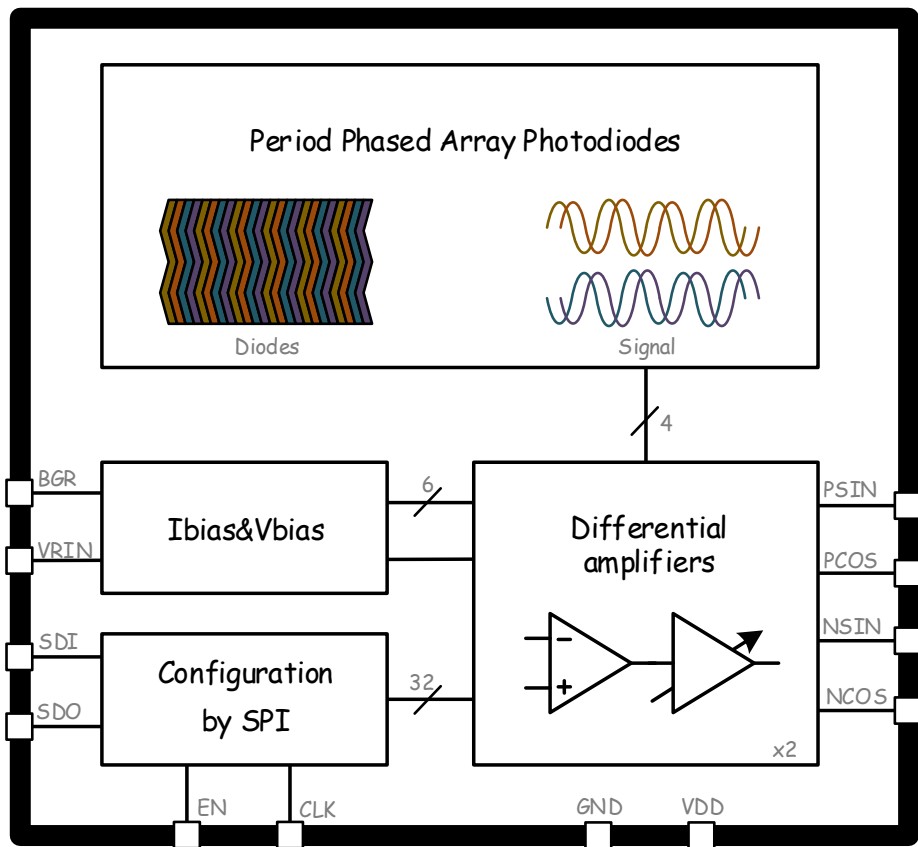

**Figure 2.** Framework of the phased array detector chip.

## 3. Design of the Period Phased Array Photodiodes

As shown in Figure 3, we analyzed four different detector arrangements. The period of the code wheel is 88.9 μm, which is significantly larger than the wavelength of the light source, i.e., 650 nm. Therefore, it can be analyzed by the principle of shading in geometric optics. The generated photocurrent is represented by the overlapping region between the light transmission of the code wheel and the detector without considering the diffraction of the code wheel [20,21]. We use the model to simulate on MATLAB software and obtain the photocurrent waveform generated by the detector. The simulation simulates the overlapping region between the light transmission and the detector when the code disk is moving laterally. The working state of the detector array is simulated by the model of the overlapping region change between the code wheel and the detector when the code wheel is in horizontal motion.

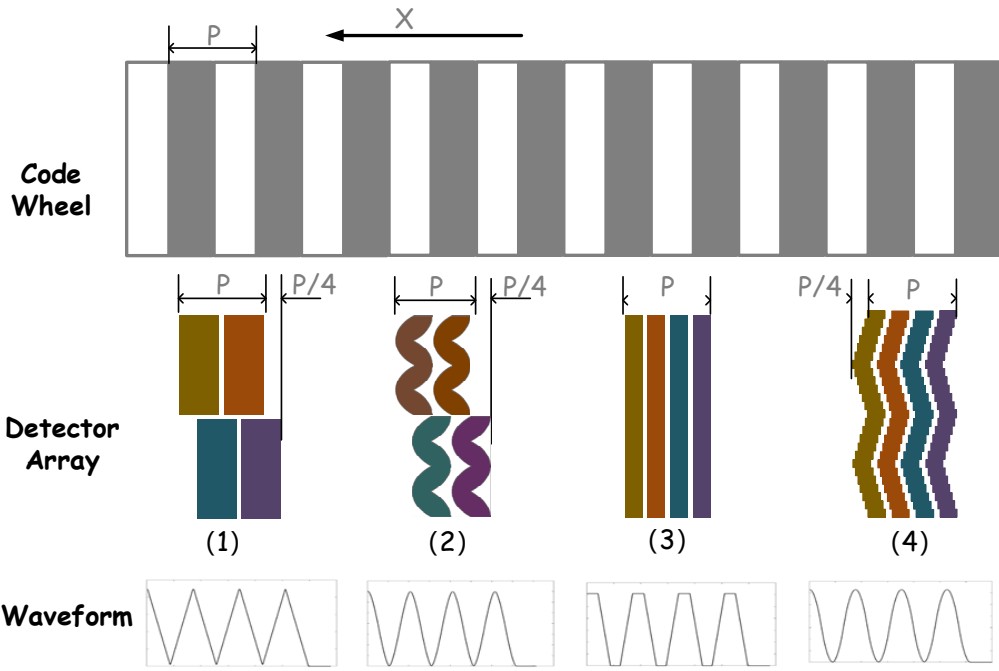

**Figure 3.** Arrangements of different phased arrays.

As shown in Figure 3, the signal received by the detector in Scheme (1) and Scheme (2) presents a triangular waveform. However, the shape of the detector is improved so that the received signal presents a waveform similar to a sine wave. Scheme (3) and Scheme (4) are phase array detectors, which can clearly observe the change in waveform.

According to the above research phenomenon, we use a mathematical modeling method to analyze the influence of detector shape on waveform. The signal generated by the detector is periodic and meets the Dirichlet condition, so we can express the signal waveform with Fourier expansion. Take Scheme (3) as an example; the periodic trapezoidal wave is expressed as follows:

$$f_{(x)} = \begin{cases} \frac{4x}{T}; & 0 < x \le \frac{T}{4} \\ 1; & \frac{T}{4} < x \le \frac{T}{2} \\ 1 - \frac{4x}{T}; & \frac{T}{2} < x \le \frac{3T}{4} \\ 0; & \frac{3T}{4} < x \le T \end{cases} \tag{1}$$

$f_{(x)}$ is expanded into a Fourier series as follows:

$$f_{(x)} = a_0 + \sum_{n=1}^{\infty} (a_n \cos nx + b_n \sin nx) = a_0 + \sum_{n=1}^{\infty} A_n \sin(nx + \varphi_n) \tag{2}$$

In the above formula, $a_0$ can be used to characterize the drift of the DC level. $A_1 \ldots A_n$ is the amplitude of each harmonic in a signal, which can be used to characterize the constant amplitude of a multiphase signal. Further, $1 \ldots n$ is the phase angle of each harmonic, which can be used to characterize the orthogonality of multiphase signals [22].

We regard the detector in Scheme (4) as a group of misplaced pixels, and the Fourier expansion of each pixel can be expressed as:

$$f_{m(x)} = a_0 + \sum_{n=1}^{\infty} A_n \sin\left(nx + \varphi_n + \frac{m\pi}{m-1}\right) \tag{3}$$

The *m* in the above formula represents the number of functions of different phase relations. By superimposing multiple Fourier expansions of different phase relations, we can express the signal in Scheme (4) as follows:

$$f_{(x)} = \sigma \sum_{m=1}^{\infty} f_{m(x)} \tag{4}$$

This equation is computed in MATLAB to obtain the waveform of $f_{(x)}$. Figure 4 shows the harmonic component of $f_{(x)}$ when the m is different. The red data (when the number of detectors $m = 1$) show the spectrum without phase difference filtering. By comparing the green data, we can find that the third and fifth harmonic components can be effectively reduced by superposing the detectors arranged in phased array. By further comparing the blue data, we can conclude that, when the number of detectors stacked in the phase array is larger, the phase difference of each detector is smaller, and the suppression effect on harmonic components is more obvious so that the sine of the signal waveform is better. According to the above discussion, we have designed the "broken line" period phased array photodiodes in this article, and the layout will be reflected in the following chapters.

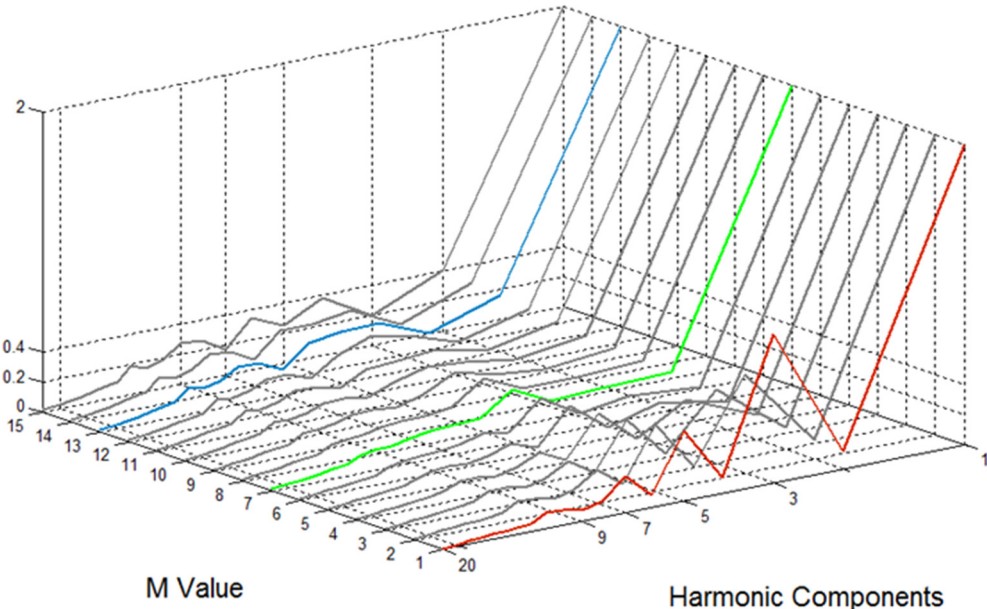

**Figure 4.** The effect of the number of detectors on harmonic components.

## 4. Design of the ASIC Chip

As shown in Figure 5, the main circuit architecture is composed of transimpedance amplifier (TIA), full differential amplifier (AMP), resistance array (RX8 and RTI), bias circuit and some control switches. The amplifier circuit adopts a two-level operational amplifier architecture. Considering the possible problems caused by the difference in actual light intensity, a multi-gear adjustment range is reserved. The amplifier gain can be adjusted to a certain extent by adjusting the resistance value. In addition, bandwidth and noise are also the main indicators to be considered.

Regarding the circuit structure of a differential amplifier, its input stage uses the cascode structure, which can effectively isolate the stray capacitance of the input and meet the bandwidth requirements; at the same time, in order to meet the gain requirements of TIA, the second stage amplifier with a load is used as the output stage. On the premise of ensuring the above indicators, the noise of TIA should be as low as possible because the current detected by the detector is only a few hundred nanoamps [23]. Considering the problems that the actual light intensity difference may bring, the range of multi-level

adjustment is reserved. By adjusting the resistance value, the gain of the amplifier can be adjusted to a certain extent.

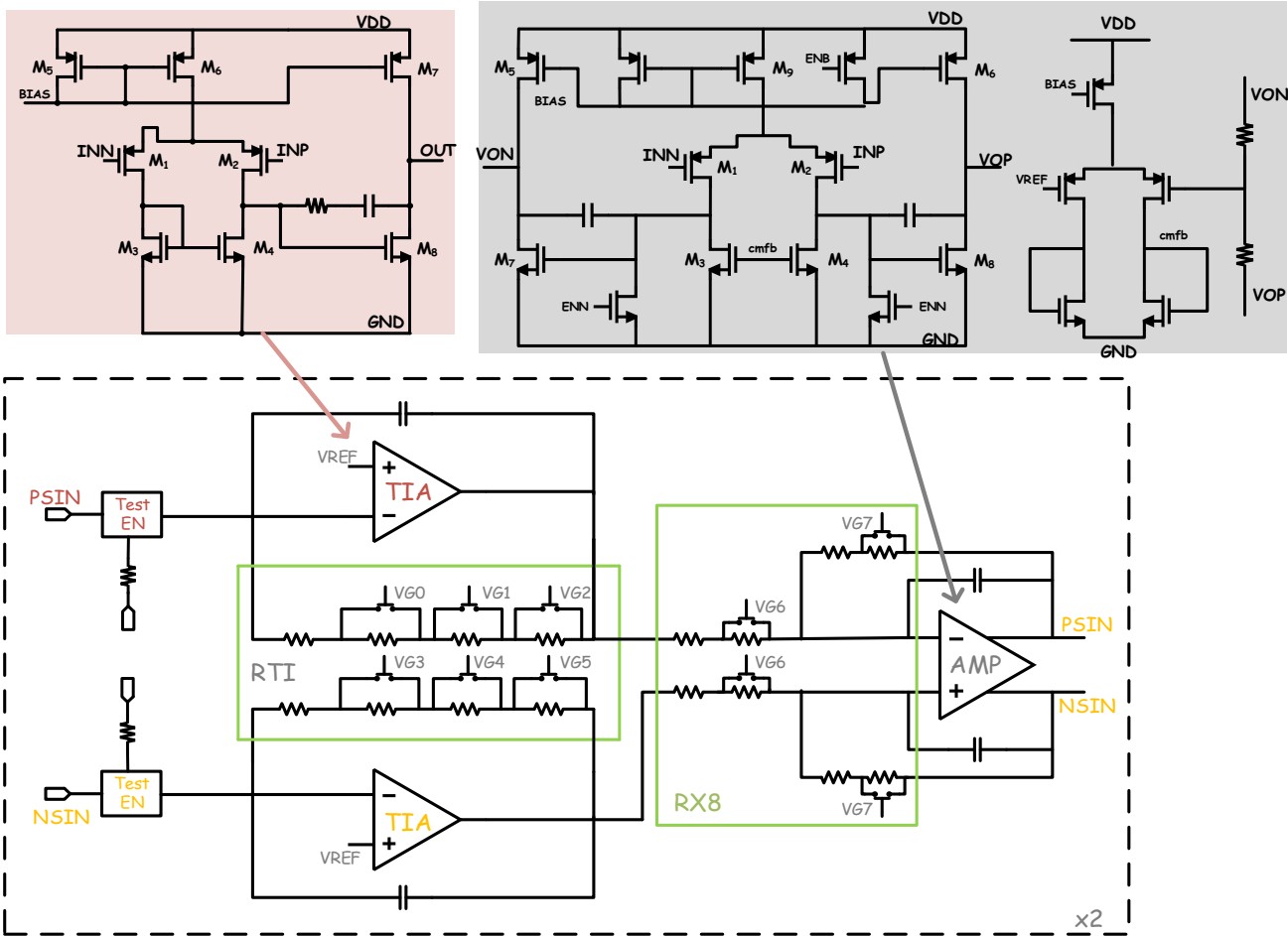

**Figure 5.** Circuit architecture of operational amplifier: TIA architecture is shown in dark red; The architecture of AMP is shown in grey; Resistance array RX8 and RTI are selected from the box.

Regarding the structure of the fully differential amplifier, in order to meet the design requirements of low power consumption and low noise, the differential pair transistor is selected as the input stage and the load transistor and Miller compensation capacitor are added; the current bias is introduced by the gates of the mirror current sources M5, M6, M9 and provides current for each branch. According to the current formula of the MOS transistor in the saturation region, the aspect ratio of each transistor in the circuit can be calculated. In addition, considering that the main sources of noise in the circuit are MOS transistor thermal noise and $1/f$ noise, PMOS is used as the input differential pair transistor of the amplifier because the carrier mobility of PMOS transistors is several times larger than that of NMOS transistors, the $1/f$ noise of PMOS transistors is several times smaller than that of NMOS transistors and the noise of other PMOS transistors as current sources can be ignored.

In order to better reduce the impact of parasitic parameters and layout mismatch, deep well technology and isolation design are added to the layout design. The MOSFET with larger size in the circuit adopts the way of cross connection, and the wiring is as symmetrical as possible. In order to ensure the consistency of the two amplifier circuits, not only is the amplifier matched but the feedback resistor of the amplifier is designed by array. In addition, the components of capacitor and resistor are isolated by dummy. The layout and size of some key modules are shown in Figure 6.

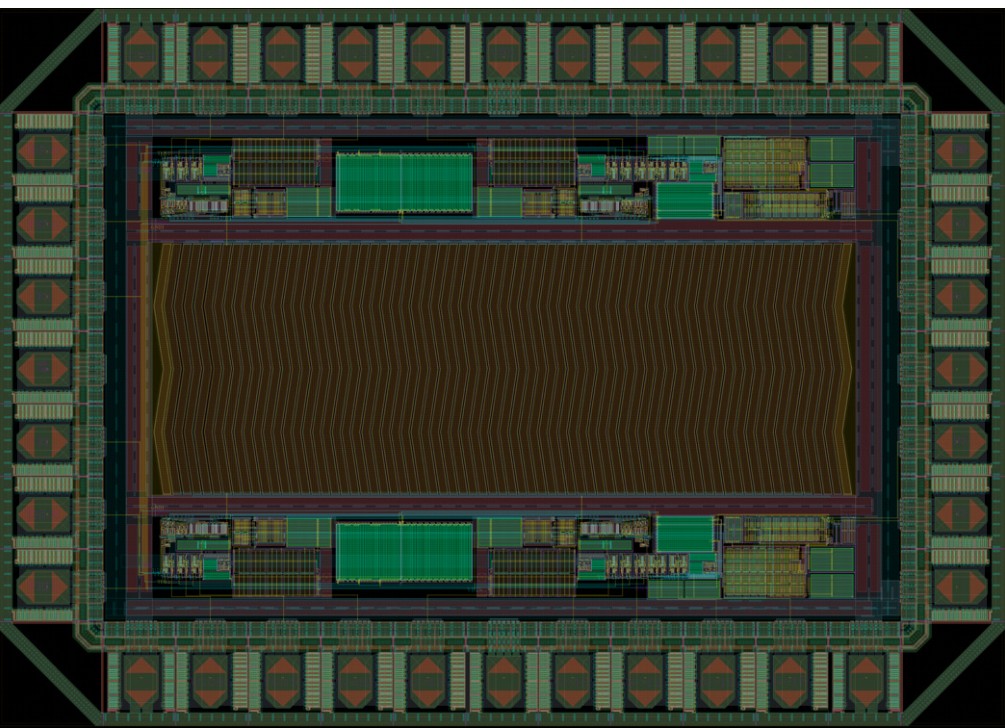

**Figure 6.** Layout of the phased array detector chip.

## 5. Verification and Simulation

In the detector design, we mainly focus on dark current and isolation. Based on the previous research results, if we compress the detector size small enough, we can obtain a better sinusoidal signal. However, this method also results in layout difficulty and noise impact. We designed the detector to be "broken lines", which not only avoids the complicated connection between detectors but also improves the sinusoidal character of the signal to the maximum extent.

Next, referring to the photoelectric responsiveness index in PDK (the corresponding response rate of 670 nm is 0.44), through the window area of the detector, we can calculate that the order of photocurrent is 10 nA, the order of dark current is 1 pF and the capacitance is 0.7848 pF. According to the model structure and parameters mentioned above, the isolation degree is simulated in TCAD simulation software.

In this model, there are two identical and adjacent photo diodes, one of which is added with a light source, and then the photocurrent of the two photo diodes is detected. As shown in Figure 7, it can be found that the current difference between two adjacent photodiodes is more than 40 dB, which means that the devices can achieve good isolation.

The layout of the amplifier circuit above is simulated. Four ways sine and cosine current signals with a phase difference of 90° are used to replace the input of the detector (the frequency is 79 k; current peak-to-peak value is 200 nA). The magnification of the first stage is set to 522 kΩ, the magnification of the second stage is set to four times and the output load is 20 pF.

The transient simulation results are shown in Figure 8. The abscissa represents time, and the ordinate represents the voltage value of the output signal. The phase and amplitude of the four-channel output signal are consistent with the theoretical value, and the circuit has realized the basic functions. It can be seen from the figure that the center of the fitted Lissajou circle of the two outputs is (1.25, 1.25), that is, the off-chip reference voltage VRIN. The diameter of the circle corresponds to the VPP of the output signal.

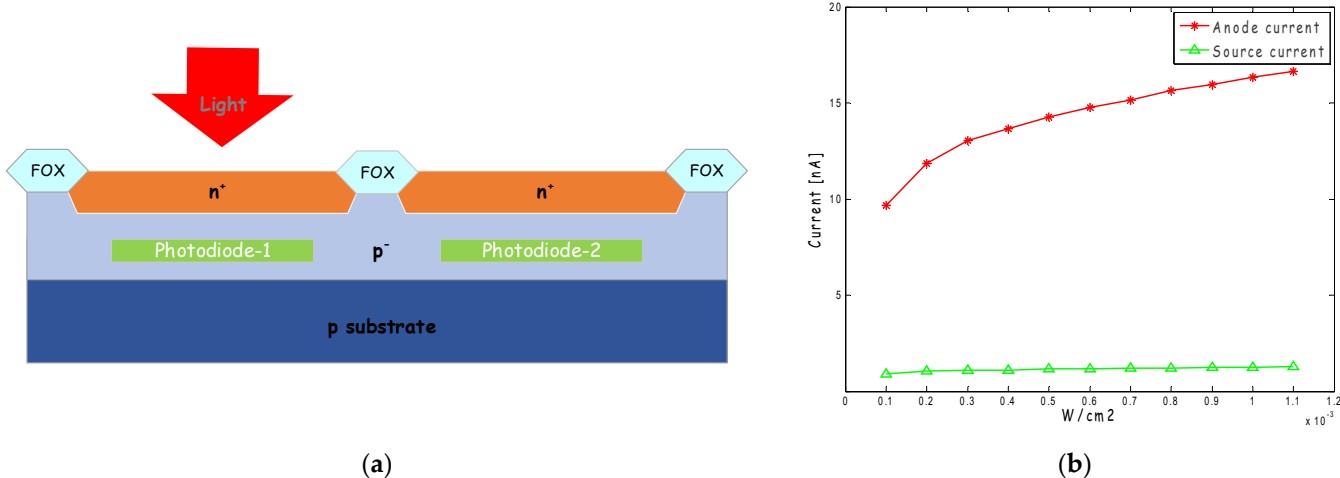

(**a**)

(**b**)

**Figure 7.** Simulation results of photodetector: (**a**) is the simulation modeling of detector; (**b**) is the simulation result of isolation.

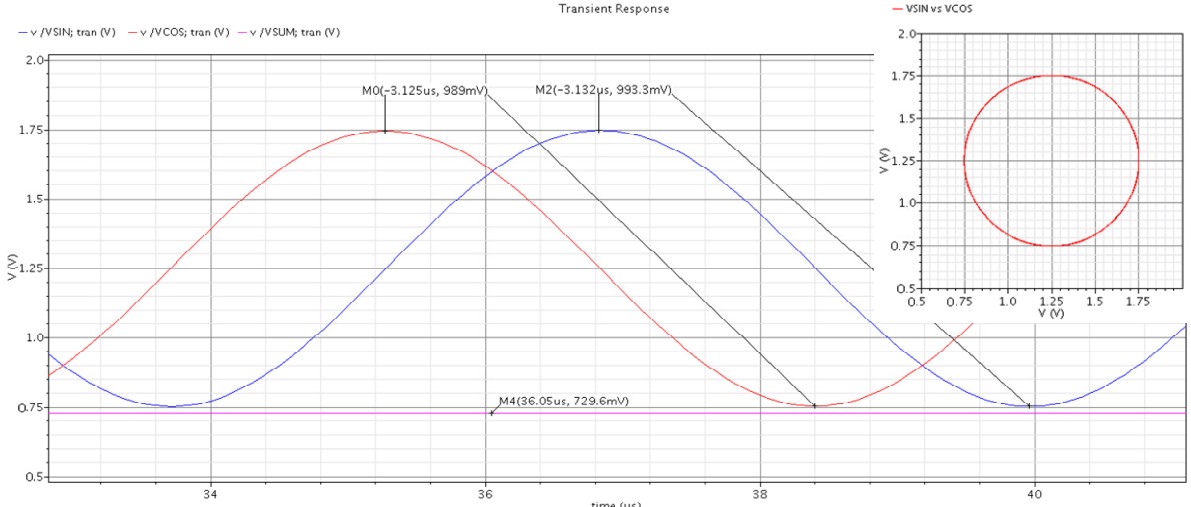

**Figure 8.** Simulation results of transient.

The AC characteristic of the operational amplifier is simulated by Monte Carlo for 50 times. In the Monte Carlo simulation, the process mismatch and corner cases are considered, and the Monte Carlo simulation parameters are set. The result is shown in Figure 9. Comparison of any two results shows that the difference is very small, and the frequency is stable at 490 k Hz, with the effective frequency range of the system less than −3 dB.

The noise of the output signal is simulated, the output signal in the figure above is transformed by FFT, the sampling time is set to 5.122 ms and the sampling frequency is set to 1 M Hz. The result is shown in Figure 10. Among them, ENOB reached 10.5 bit, SNR reached 60.92 dB and SNDR was 79.63 dB.

Furthermore, the corner simulation under different temperature conditions was performed for the photodiodes and MOS devices. The simulation results are listed in Table 1. For various cases, the SNRs of all output signals are greater than 60.92 dB.

In summary, the response frequency of the system reaches 490 k, which can meet the requirements of all photoelectric encoders. In addition, the output signal has good orthogonality, and the SNRs are greater than 60 dB.

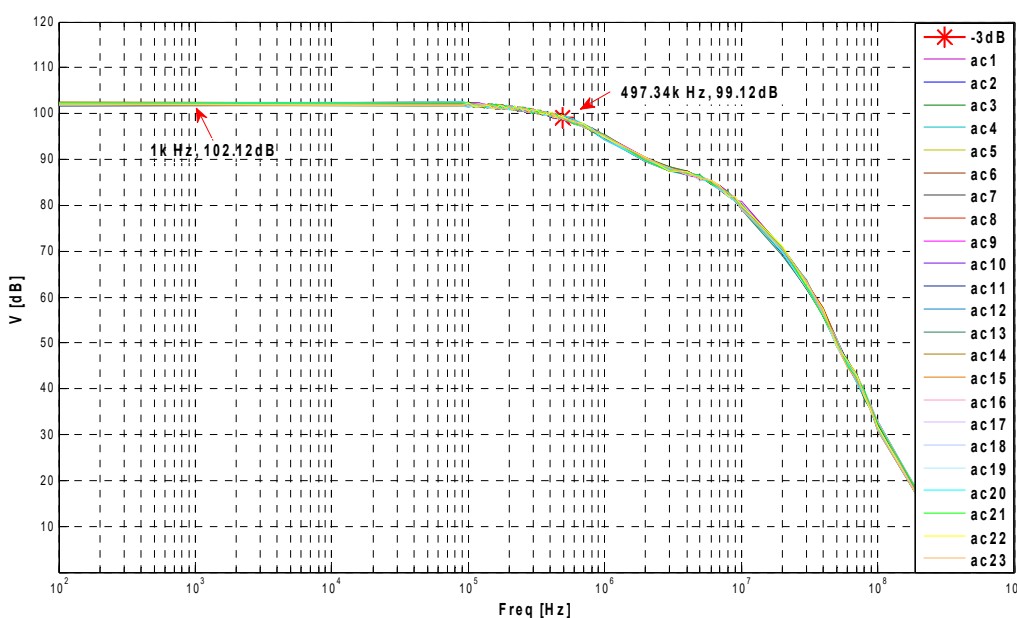

**Figure 9.** Monte Carlo simulation of AC characteristics.

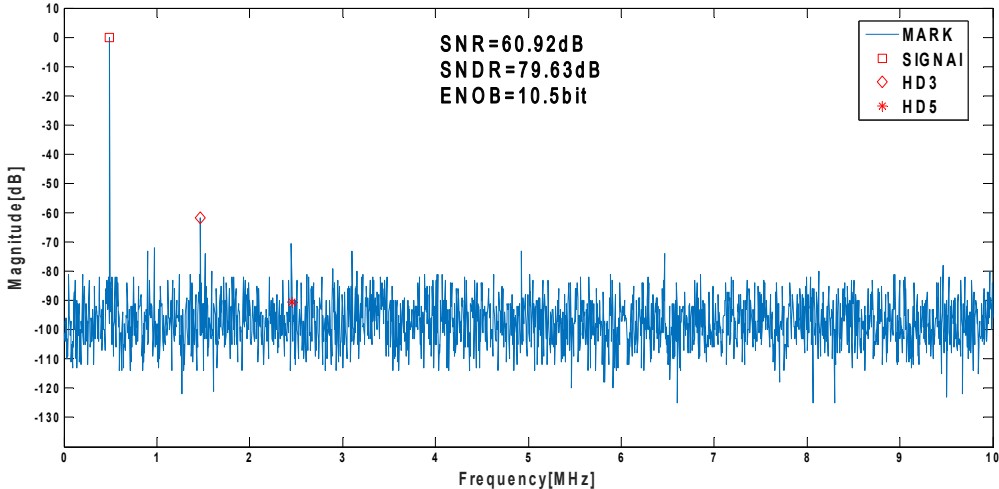

**Figure 10.** Simulation results of noise.

**Table 1.** Simulation results of each corner component.

|  | **−40 °C** | **27 °C** | **125 °C** |
|---|---|---|---|
| Photodiodes: tm/wp/ws | 62.31 dB | 61.92 dB | 61.52 dB |
| MOS: tm/wp/ws/wo/wz | 62.31 dB | 61.92 dB | 61.52 dB |
| Capacitance: tm | 62.31 dB | 61.92 dB | 61.52 dB |
| Capacitance: wp | 61.75 dB | 61.39 dB | 60.97 dB |
| Capacitance: ws | 62.75 dB | 62.34 dB | 61.96 dB |
| Resistance: tm | 62.31 dB | 61.92 dB | 61.52 dB |
| Resistance: wp | 60.94 dB | 60.60 dB | 60.26 dB |
| Resistance: ws | 63.41 dB | 63.02 dB | 62.60 dB |

## 6. Noise Optimization

The noise in the designed phased array detection chip is mainly attributed to the dark current and the amplified circuit of the optoelectronic detector. The dark current of the detector is a type of shot noise, which is induced by the surface defects or body defects in the optoelectronic receiving surface during the chip manufacturing process. To ensure

the performance of the detector, the standard photodiode model in the database of XFAB technology is selected for designing the detector array.

Figure 11 shows the structure profile of the high-speed p-i-n photo diode and optical window etching, with anti-reflective coating (ARC) deposition for photo diode protection and sensitivity improvement. When available as part of the final passivation layer, silicon nitride is used as an ARC in X-FAB technology families with a well index of refraction, and sufficient protection of the photo diode and metallization system by the thin ARC layer has been demonstrated, excluding any reliability issue due to integration of the opto-window process steps to the CMOS process. According to the simulation data in the previous chapter, we obtain the dark current value per unit area as 0.75 fA/$\mu$m$^2$, the optoelectronic response rate as 0.44 A/W and the input light intensity as 0.2 mW/cm$^2$, such that the reference noise of the photocurrent output is 13.02 pA (at capacitance 4.608 pf and 400 kHz). In this study, because the dark current is a statistical value [24], multiple photo diodes in series reduce the value of the dark current, which is less than the theoretical value.

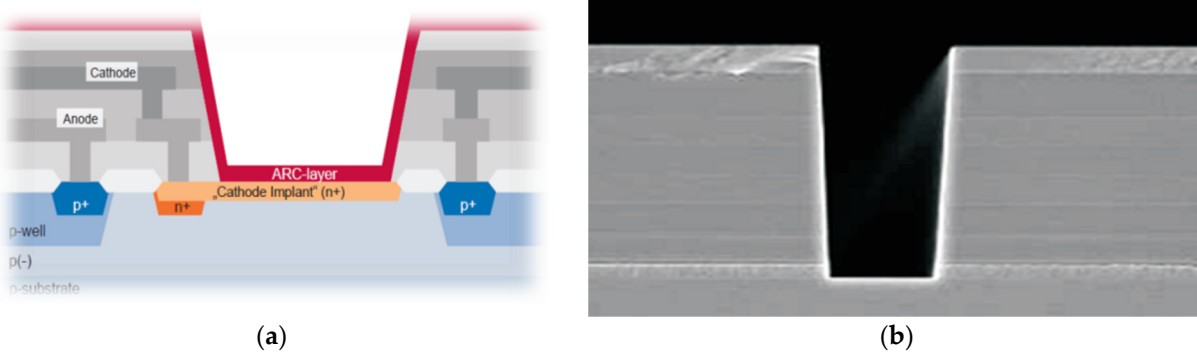

(**a**) (**b**)

**Figure 11.** High-speed p-i-n photodiode structure in X-FAB process library: (**a**) section of the structure; (**b**) optical window etching.

The design of operational amplifier adopts traditional architecture, which has relatively low power consumption and noise. The main sources of noise are MOS transistor thermal noise and $1/f$ noise in amplifier. Thermal noise can be suppressed by increasing gm, and $1/f$ noise can be reduced by increasing MOS transistor area. In the actual design, a single MOS transistor in TIA is designed as a dactylitis texture. When it works in the saturated state, the channel noise of the source electrode and drain of MOS transistor can be expressed as:

$$i^2_{n(ch)}(f) = 4kT\gamma g_m \tag{5}$$

where $k$ is the Boltzmann constant, $T$ is the Kelvin temperature and gm is the transconductance of MOSFET. For long channel MOSFET, the coefficient $\gamma$ is 0.6. The resistance of metal oxide in the source, gate and drain materials of CMOS transistors can also cause noise. The resistance $R_g$ of the gate cannot be ignored; the thermal noise is as follows:

$$v^2_{n(R_g)}(f) = \frac{4kTR_g}{3} \tag{6}$$

Flicker noise is caused by random capture and release of charge carriers near the diffusion layer of silicon-based materials. The noise is:

$$v^2_{n(1/f)}(f) = \frac{K}{c_{ox}WL} \times \frac{1}{f} \tag{7}$$

where $K_f$ is the flicker noise figure, $C_{ox}$ is the gate capacitance, $W$ and $L$ are the length and width of MOSFET. According to the above formula, the values of all thermal noise and scintillation noise are about 200 fA·Hz$^{-1/2}$; this part of the noise current cannot be

eliminated and amplified from the input to the output [25,26]. Therefore, in the design of TIA, decoupling capacitor should be added to limit.

In addition, PMOS is used as input differential pair transistor of amplifier. Because the carrier mobility of PMOS transistor is about 2–5 times larger than NMOS transistor, the $1/f$ noise of PMOS transistor is 2–5 times smaller than NMOS. The noise of other PMOS transistors as current source is negligible; the noise of the input stages M1 and $M_3$ should be multiplied by the gain of the second stage amplifier, respectively. The noise of $M_2$ and $M_4$ is equal to that of $M_1$ and $M_3$, and the formula is expressed as a double relation. The noise of the output stage is negligible because it is not amplified. The total output noise formula is:

$$\overline{V_{noise,out}^2} = 2g_{m8}^2 r_{o8}^2 \left[ 4kT\frac{2}{3}(g_{m1}+g_{m3})(r_{o1}\|r_{o3})^2 + \frac{K}{C_{ox}W_1L_1}\frac{1}{f}g_{m1}^2(r_{o1}\|r_{o3})^2 + \frac{K}{C_{ox}W_3L_3}\frac{1}{f}g_{m3}^2(r_{o1}\|r_{o3})^2 \right] \quad (8)$$

The equivalent input noise at the input is:

$$\overline{V_{noise,in}^2} \gg \frac{\overline{V_{noise,out}^2}}{A_v^2} = 2\left[ \frac{8}{3}kT\left(\frac{1}{g_{m1}}+\frac{g_{m3}}{g_{m1}^2}\right) + \frac{K}{C_{ox}W_1L_1}\frac{1}{f} + \frac{K}{C_{ox}W_3L_3}\frac{1}{f}\frac{g_{m3}^2}{g_{m1}^2} \right] \quad (9)$$

The capacitance in this design is, in total, 210 fF, which is brought into the formula, and the total noise of the chip is below $1000e^-$ magnitude. Figure 12 shows the proportion of calculated results of each noise. The size of the detection is large, and the dark current is 13.02 pA according to the dark current value of 0.75 fA/um$^2$ per unit area. The detector's dark current far exceeds the noise of the circuit, so the key of chip design is to ensure the uniformity of the detector.

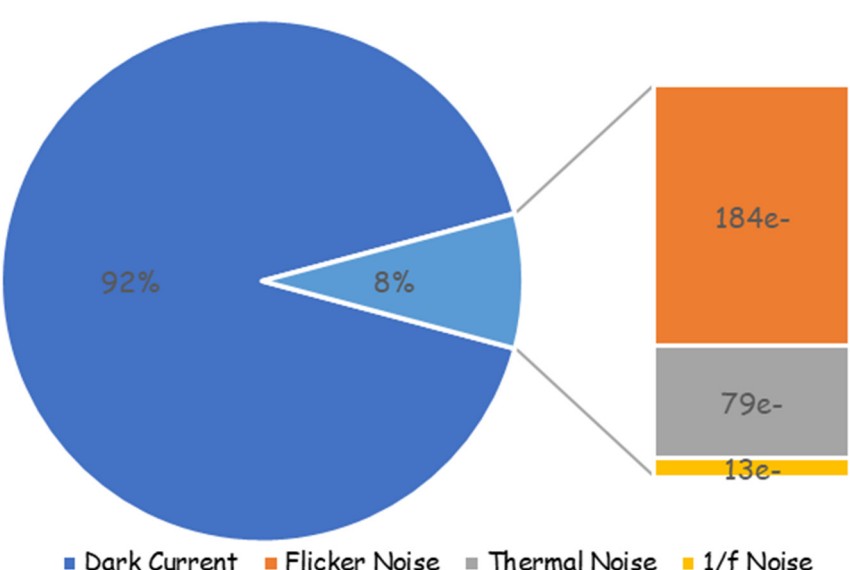

**Figure 12.** Composition and proportion of main noise.

As shown in Table 2, we compared three products of iC-Haus, which represent the highest technology and process level in the industry. The chip in this paper is different from two traditional products. On the premise of ensuring the main indicators, it can output a sine signal, which can provide higher resolution of the encoder and be processed by a digital circuit. In addition, due to the unique design of the detector, the chip designed in this paper can obtain an output signal with a high signal-to-noise ratio without complicated processing circuits. Therefore, it has obvious advantages in two key indicators: die size and SNR, which reflects the advantages of this design scheme.

**Table 2.** This is a table. Tables should be placed in the main text near to the first time they are cited.

|  | iC-LSHB | iC-PN | iC-LG | This Paper |
|---|---|---|---|---|
| Output signal | Square wave | Square wave | Sine wave | Sine wave |
| Die size | 2.9 mm × 2 mm | 6.2 mm × 5.2 mm | 3.5 mm × 7 mm | 2 mm × 1.8 mm |
| Optoelectronic response rate | 0.25 A/W | 0.5 A/W | / | 0.44 A/W |
| Response frequency | 400 kHz | 400 kHz | 500 kHz | 400 kHz |
| Signal-to-noise ratio | / | / | 8 bit | >60 dB (10 bit) |

## 7. Conclusions

This paper presents a novel phased array optoelectronic detection array that applies the principle of phase difference filtering to the design of the photodetector. The chip design can improve the system accuracy and integration of the photoelectric encoder system, which is attributed to the "broken line" design not only playing a role in reducing harmonics but also meeting the needs of a compact array layout.

Further, we completed the ASIC design to verify the feasibility of the ideas in this paper. The main work includes verification of the performance and process of the "broken line" detector, and the design of the appropriate circuit system. The simulation results show that this chip outperforms the same type of products in terms of photoelectric efficiency, signal quality, response frequency and other technical indicators.

**Author Contributions:** Conceptualization, Y.C.; methodology, Y.M.; validation, N.H., C.W. and Y.Z.; investigation, K.C.; writing—original draft preparation, Y.M. and N.H.; writing—review and editing, Y.M. All authors have read and agreed to the published version of the manuscript.

**Funding:** This work was supported by the Scientific Research Project of Education Department of Jilin Province under Grant No. JJKH20220262KJ and JJKH20210276KJ.

**Institutional Review Board Statement:** Not applicable.

**Informed Consent Statement:** Not applicable.

**Data Availability Statement:** Not applicable.

**Conflicts of Interest:** The authors declare no conflict of interest.

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
