# Peer review of "An Optoelectronic Detector with High Precision for Compact Grating Encoder Application"

_electronics, doi:10.3390/electronics11213486_

Round 1

Reviewer 1 Report

The document presents applied research; however, it does not clearly show innovation compared to other publications between 2023-2019.

It is suggested to the authors:

* Change the structure of document 1.- Introduction (Generalities of the problem using references between 2023 and 2019); 2.- Related Works (Proposed solutions to the problem stated in the abstract using references between 2023-2019); 3.- Problem Formulation and Methodology (Table of variables of the mathematical model, mathematical model, pseudocode, flowchart of the implemented methodology); 4.- Analysis of Results (Metrics that demonstrate the performance of the proposal. Do not use screenshots of simulation software. Use only metrics that can perform directly in Overleaf or Matlab); 5.- Conclusions (Relate the results found with the objectives stated in summary to solve the problem. Without a problem there is no research); 6.- References (Use current references between 2023 and 2019. Use Sciencedirect, MDPI, Wiley, Taylor & Francis, Hindawi, SAGE, Springer, IEEE [Transactions, Conferences, Journals]).

The quality of the figures is very low for a scientific article; it is suggested to use Overleaf as a word processor and the figures in PDF format to guarantee the quality. Figure 6 has inadequate gray borders.

Keep the same font in the figures.

Author Response

Dear

Thanks to the reviewer's professional and patient guidance, this article has been significantly improved. I have studied reviewer’s comments carefully and have made revision which marked in the revised manuscript. The main issues concerned by reviewers are revised as follows:

Q1. Article does not clearly show innovation compared to other publications between 2023-2019.

The innovation in this paper is mainly embodied in the design of the broken line detector array. Through mathematical modeling and simulation, we have proved the advantages of the broken line detector array in receiving and restoring signals. In order to verify the feasibility of this scheme, we completed the SoC, which has a complete and professional IC design process, and the simulation and analysis results of key indicators are also higher than those of similar products.

In conclusion, I made targeted modifications in the abstract, conclusion and other key chapters according to your suggestions. Several major changes include: “A simulation model of the "broken line" detector structure and process was established, which only meets the needs of compact array layout,but also ensure a good photoelectric conversion rate. In addition, we used a professional design program to complete the layout of the ASIC, which maximized the recovery of the signal received by the detector. The simulation and noise analysis results show that the SNRs of the output signal are greater than 60 dB with 400kHz response frequency.”(Located in 14 line with word manuscript).“The chip designed can improve the system accuracy and integration of the photoelectric encoder system, which is attributed to the "broken line" design not only plays a role in reducing harmonics, but also meets the needs of compact array layout” (Located in 294 line with word manuscript). “Further, we completed the ASIC design to verify the feasibility of the ideas in this paper. The main work includes the verification of the performance and process of the "broken line" detector, and the design of the appropriate circuit system. The simulation results show that this chip outperforms the same type of products in terms of photoelectric efficiency, Signal quality, response frequency and other technical indicators” (Located in 348 line with word manuscript). et al.

New and replaced references are as follows:

[1] Das, S.; Chakraborty, B. Design and Realization of an Optical Rotary Sensor [J]. IEEE Sens. J. 2018, Vol. 18, pp2675–2681.

[9] Jiaqi Jiang, Hongbo Zhang, Yunhao Fu, Yuchun Chang. A Phased-array Optoelectronic Detector using Phase-difference Filtering Technology for Incremental Encoder Application[C]. 2021 IEEE 14th International Conference on ASIC (ASICON), 2021, Vol. 2021, pp1-4.

[11] Han Hou, Guohua Cao, Hongchang Ding, Kun Li. Research on Particle Swarm Compensation Method for Subdivision Error Optimization of Photoelectric Encoder Based on Parallel Iteration[J]. Sensors, 2022 22(12), pp4456.

[12] Guoyong Ye, Zeze Wu, Zhengchen Xu, YangWang Yongsheng Shi, Hongzhong Liu. Development of a digital interpolation module for high-resolution sinusoidal encoders [J]. Sensors and Actuators A: Physical, 2019, Vol. 285, pp501-510.

[16] Yi Sun, Yan Tian, Yiping Xu. Problems of encoder-decoder frameworks for high-resolution remote sensing image segmentation: Structural stereotype and insufficient learning[J]. Neurocomputing, 2019, Vol. 330, pp297-304.

[17] Jiaqi Jiang, Jiahai Dai, Shang Yang, Yunchun Chang. A22-bit image encoder with optoelectronic integrated chip[J]. Optics Communications, 2022, Vol. 512, 128022.

[20] Jun-Yan Liu, Yang Liu, You Wu, Yang Wang. Characterization of ion radiation-induced damage in polyimide by wavelength-scanning and dual-wavelength laser photoacoustic spectroscopy[J]. Polymer, 2019, Vol. 165, pp55-60.

[21] Ci Sun, Mingjia Wang, Jicheng Cui, Xuefeng Yao, Jianjun Chen. Comparison and analysis of wavelength calibration methods for prism – Grating imaging spectrometer[J]. Results in Physics, 2019, Vol. 12, pp143-146.

[22] Jing Qin, R. M. Silver, et al. Deep subwavelength nanometric image reconstruction using Fourier domain optical normalization[J]. Light Science &Applications, 2016, Vol. 5, e16038

[23]Cheng Xiang, Zhu Li-Jun, et al. Development of multi-function digital optoelectronic integrated sensor[J]. Optik, 2019, Vol. 180, pp406-413.

Q2. Change the structure of document. 

With reference to your suggestions, I have made some adjustments to the structure of this paper. The revised chapters are as follows:1. Introduction (Generalities of the problem);2. Design of the phased array detector chip(Proposed overall plan in the abstract)ï¼›3. Design of the Period Phased Array Photodiodes(Theoretical Modeling and Scientific Hypothesis)ï¼›4. Design of the ASIC chip (Related Works);5. Verification and simulation (Simulation analysis and verification);6. Noise optimization (Analysis of Results);7. Conclusions (Results and hypothesis validation).

Q3. The quality of the figures is very low(especially Figure 6),and keep the same font in the figures.

In the revised manuscript, I entered and replaced Figures 3, 6, 9 and 10 in the original. I also uploaded a PDF version of the revised manuscript to ensure the clarity of the picture.

Reviewer 2 Report

This article tries to present a  novel optoelectronic detection array, which adopts the research idea of op-tical, mechanical and electrical integration with >60-dB SNR and 400-kHz response frequency. The article is not well organized and lacks novelty and technical contributions. Some essential concerns need to be addressed:

(1) The novelty and significance of the proposed optoelectronic detection array need to be highlighted.

(2) The presented results are all from simulation. Are there prototype verifications?

(3) The comparison table with state-of-the-art designs in literatures are needed.

(4) The figure quality is poor and hard to tell the key information.

(5) The structure and organization of the article is not clear.

(6) Have no idea why screenshots are presented in a paper, e.g. Figs. 9 and 10.

Author Response

Dear

Thanks to the reviewer's professional and patient guidance, this article has been significantly improved. I have studied reviewer’s comments carefully and have made revision which marked in the revised manuscript. The main issues concerned by reviewers are revised as follows:

Q1. The novelty and significance of the proposed optoelectronic detection array need to be highlighted.

The innovation in this paper is mainly embodied in the design of the broken line detector array. Through mathematical modeling and simulation, we have proved the advantages of the broken line detector array in receiving and restoring signals. In order to verify the feasibility of this scheme, we completed the SoC, which has a complete and professional IC design process, and the simulation and analysis results of key indicators are also higher than those of similar products.

In conclusion, I made targeted modifications in the abstract, conclusion and other key chapters according to your suggestions. Several major changes include: “A simulation model of the "broken line" detector structure and process was established, which only meets the needs of compact array layout,but also ensure a good photoelectric conversion rate. In addition, we used a professional design program to complete the layout of the ASIC, which maximized the recovery of the signal received by the detector. The simulation and noise analysis results show that the SNRs of the output signal are greater than 60 dB with 400kHz response frequency.”(Located in 14 line with word manuscript).“The chip designed can improve the system accuracy and integration of the photoelectric encoder system, which is attributed to the "broken line" design not only plays a role in reducing harmonics, but also meets the needs of compact array layout” (Located in 294 line with word manuscript). “Further, we completed the ASIC design to verify the feasibility of the ideas in this paper. The main work includes the verification of the performance and process of the "broken line" detector, and the design of the appropriate circuit system. The simulation results show that this chip outperforms the same type of products in terms of photoelectric efficiency, Signal quality, response frequency and other technical indicators” (Located in 348 line with word manuscript). et al.

Q2. The presented results are all from simulation. Are there prototype verifications?

Unfortunately, the chip designed in this paper has not been verified by Tape-out. However, the photodiodes design has been verified by mathematical model analysis and TCAD software simulation, and the ASIC chip design process is also perfect before Tape-out.

Q3. The comparison table with state-of-the-art designs in literatures are needed.

The comparison table is in the sixth chapter of the text at line 290. It is necessary for me to explain this to you, because the chip designed in this article is only used for grating encoder, and the research in literature is not suitable for comparison with it. Therefore, we compare the chip in this paper with the products of iC-Haus company, demonstrate the superiority over their predicated approach with Table 2.

Q4. The figure quality is poor and hard to tell the key information.

I'm sorry that my mistakes have brought inconvenience to your review. In the revised manuscript, I replaced all figures and highlighted key information. I also uploaded a PDF version of the revised manuscript to ensure the clarity of the figures.

Q5. The structure and organization of the article is not clear.

With reference to your suggestions, I have made some adjustments to the structure of this paper. The revised chapters are as follows:1. Introduction (Generalities of the problem);2. Design of the phased array detector chip(Proposed overall plan in the abstract)ï¼›3. Design of the Period Phased Array Photodiodes(Theoretical Modeling and Scientific Hypothesis)ï¼›4. Design of the ASIC chip (Related Works);5. Verification and simulation (Simulation analysis and verification);6. Noise optimization (Analysis of Results);7. Conclusions (Results and hypothesis validation).

Q6. Have no idea why screenshots are presented in a paper, e.g. Figs. 9 and 10.

I'm sorry. I have corrected this problem in the revised manuscript.

We have tried our best to revise our manuscript according to the comment. Once again, thank you very much for your comments and suggest.

Reviewer 3 Report

The paper contains a proper study of the documentation and the existing limitations, it tries to solve the problem of compromise between high accuracy and miniaturization through a simulation regardin the optical, electrical and mechanical  integration. Hopefully in the future, the study can be also practically verified. It seems, through simulation, that the issues that appeared at the beginnng were solved with the proposed model.

The results, formulas and figures are clearly presented, maybe in fig.9 the values are not visible, but if they are indicated during the explanations, it is ok.

Author Response

Dear

Thanks to the reviewer's professional guidance and  recognition, this article has been significantly improved. I have studied reviewer’s comments carefully and have made revision which marked in the revised manuscript. For reviewer's questions about fig.9, I have corrected this problem in the revised manuscript. Once again, thank you very much for your comments and suggest.

Round 2

Reviewer 1 Report

Figures 7 and 9 should improve in resolution. Zooming the image degrades the image.

Scientific papers should be incorporated between 2023-2020 and published in ScienceDirect, Wiley, Taylor & Francis, Hindawi.

Author Response

Thanks to the reviewer's professional and patient guidance again. In the revised manuscript, I modified the introduction and references according to the review recommendations, and the new vector graphics was replaced for Figure 7 and Figure 9.

Reviewer 2 Report

The concerns are not well addressed. The motivation, novelty, and significance are not clear. Also, still I see the direct screenshot from Cadence. For an article, it is not suggested as this is not an experimental report.

Author Response

Thanks to the reviewer's professional and patient guidance again. In the revised manuscript, I have made major changes to the introductions to clarify the motivation, novelty, and significance of this article. Moreover, Figures 7 and 9 have been replaced with new vector graphics.

Round 3

Reviewer 2 Report

My concerns have been addressed and this version can be accepted.